# That’s My Cue to Eat: A Systematic Review of the Persuasiveness of Front-of-Pack Cues on Food Packages for Children vs. Adults

**DOI:** 10.3390/nu12041062

**Published:** 2020-04-11

**Authors:** Lotte Hallez, Yara Qutteina, Maxime Raedschelders, Filip Boen, Tim Smits

**Affiliations:** 1Institute for Media Studies, KU Leuven, 3000 Leuven, Belgium; lotte.hallez@kuleuven.be (L.H.); yara.qutteina@kuleuven.be (Y.Q.); maxime.raedschelders@kuleuven.be (M.R.); 2Physical Activity, Sports & Health Research Group, KU Leuven, 3001 Leuven, Belgium; filip.boen@kuleuven.be

**Keywords:** packaging, front-of-pack cues, food marketing, children, adults, eating behaviors

## Abstract

Packaging is increasingly recognized as an essential component of any marketing strategy. Visual and informational front-of-pack cues constitute salient elements of the environment that may influence what and how much someone eats. Considering their overwhelming presence on packaging of non-core foods, front-of-pack cues may contribute to the growing rates of overweight and obesity in children and adults. We conducted a systematic review to summarize the evidence concerning the impact of front-of-pack cues on choices and eating behaviors. Four electronic databases were searched for experimental studies (2009–present). This resulted in the inclusion of 57 studies (in 43 articles). We identified studies on children (3–12 years) and adults (≥ 18 years), but no studies on adolescents (12–18 years). The results suggest that children and adults are susceptible to packaging cues, with most evidence supporting the impact of visual cues. More specifically, children more often choose products with a licensed endorser and eat more from packages portraying the product with an exaggerated portion size. Adults’ eating behaviors are influenced by a range of other visual cues, mainly, package size and shape, and less so by informational cues such as labels.

## 1. Introduction

Food packages have evolved beyond their traditional role of storing and preserving and are increasingly recognized as an important marketing tool [1]. A major reason for this is that packages reach consumers at two “critical moments of truth”, namely, at purchase and at consumption [2]. People are increasingly making in-store, impulsive buying decisions. This is especially the case for low-involvement products such as food and drinks [3]. Packages also constitute an element of the food environment at the moment of consumption and can influence decisions on what to eat and how much to eat [2].

Many marketing techniques on food packages are targeted specifically at children and adolescents. Brands are increasingly devoting part of their advertising budget on marketing to younger age groups [4,5,6]. Marketers recognize the power children and adolescents have to influence purchase decisions of their families. Parents often buy brands and products their children like or ask for [7,8]. Unfortunately, these are often the same energy-dense low-nutrient foods, hereafter referred to as non-core foods, advertised to them [9]. There is growing evidence that food marketing succeeds in influencing children’s and adolescents’ eating behaviors. A meta-analysis by Boyland et al. [10] concluded that unhealthy food advertising on television and the Internet moderately increases non-core food intake in children, while there is mixed evidence on the effect of food marketing on adults [10,11,12]. This discrepancy may partly be attributed to children’s less developed advertising literacy. They do not (yet) possess the necessary skills to recognize the persuasive intent of a message, to form a disapproving attitude toward a marketing message or to use their literacy skills when needed [10,13,14]. In addition, Qutteina et al. [15] found evidence that media food marketing increases choices for and intake of non-core food among pre-adolescents (8–11 years old) and adolescents (12–18 years old). Although there is a growing body of research on the impact of advertising through traditional and online media, little is known about the impact of persuasive cues on the omnipresent food packages. This is alarming considering that marketing techniques are highly prevalent on packages of non-core food products [16,17,18]. This means that such techniques might contribute to the growing overweight and obesity rates, particularly among children and adolescents.

In today’s competitive retail environment, food products display a wide range of visual and informational front-of-pack cues [19,20,21]. Visual cues such as color, illustrations, package shape, etc. can draw attention to the product and appeal to people’s emotions. Informational cues such as labels, tables, written claims, etc. convey verbal or numerical information about the (nutritional) quality of the product and can enforce people’s cognitions [1,21]. A considerable number of studies have explored the impact of visual cues on consumption outcomes. A review by Smits et al. [22] on the impact of child-targeted endorsement concluded that endorsers affect how much children eat, and this seems to be especially the case for non-core food. Another review concluded that increased package sizes lead to increased intake, though this effect seems smaller for children than for adults [23]. People also pay more attention to and prefer transparent packages where the product is (partially) visible [24]. Similarly, a growing number of studies have investigated the impact of informational cues on cognitive outcomes and to a lesser extent on consumption outcomes. One review concluded that people find nutrition labels useful and reliable, and that these labels help them figure out healthiness of a food product [25]. Talati et al. [26] indicated in their review that front-of-pack nutrition labels can diminish the cognitive bias that arises from misleading health claims placed by marketers. Recently, more attention has been devoted to newer initiatives such as traffic light systems or the Nutri-score [27]. Studies on other types of labels such as serving size labels have found mixed effects [28].

The impact of some types of cues can be explained by normative influence. Eating behaviors are typically the result of automatic, low-involvement thinking [29]. People make quick judgements about the appropriateness of choices and portion sizes based on elements such as package design [2]. Several studies have concluded that the size of a package influences how much someone eats [23]. If one finds it appropriate to eat a specific proportion of a package, then intake increases along with size. Similarly, the depicted portion size can create expectations about how much one should eat, such that pictures where the portion size is exaggerated can lead to a higher intake [30,31]. However, other cues can potentially mitigate the effect of these (misleading) cues. A serving size label could correct the judgements people have about appropriate portion sizes, for instance, by stating that a package contains not one but multiple servings [32]. Similarly, the option to reseal a package can signal that the package contains more than one serving [33].

In other words, previous reviews have provided evidence on the impact of individual packaging cues. The aim of the current review is to summarize the experimental evidence and to provide a holistic overview of the impact of all possible front-of-pack cues on choices and food behaviors. The review includes people of all ages, and aims to draw a comparison between different age groups, specifically children, adolescents, and adults.

## 2. Methods

The current study relied on the Preferred Reporting Items for Systematic Reviews and Meta-analyses (PRISMA) guidelines [34]. The review included peer-reviewed journal articles published online between January 2009 and October 2019. The protocol for this systematic review was submitted for pre-registration in Prospero.

### 2.1. Inclusion Criteria

This review includes quantitative studies with a (quasi-)experimental design, in which at least one front-of-pack cue was experimentally manipulated and the influence on product choices, serving amounts, or consumption amounts was assessed. We defined front-of-pack cues as “all cues that are visible when standing in front of a package”. These cues had to be present on three-dimensional packages of food products or non-alcoholic beverages. The experiments could have a between-subjects or a within-subjects design, but had to allow comparison with a control condition or another intervention. Qualitative and observational studies were excluded. Participants of all ages were included in this review.

### 2.2. Data Sources and Search Strategy

The search string included key terms referring to the intervention, study design, population, and outcome of interest (see Appendix A). After piloting in Web of Science, the search string was fine-tuned and tailored for search in four electronic databases: Web of Science, PubMed, PsycARTICLES, and Google Scholar. The search yielded 6313 hits in total. After removing duplicates, 5774 articles remained for further screening. Three authors (L.H.; Y.Q.; M.R.) screened articles based on title, abstract, and full text. Two undergraduate students helped with abstract screening. The articles were checked independently by two different authors, and other members of the team were consulted in case of disagreement. The screening resulted in forty included articles. A backward search of reference lists led us to include three more articles. This resulted in a total of forty-three articles included for qualitative synthesis (see Figure 1). 

### 2.3. Data Extraction

Three authors (L.H; Y.Q.; M.R.) independently extracted data from the included articles. The extraction form contained the following fields: title, author(s), year of publication, journal, author affiliation, country/ies of author affiliation, study design, study objective, intervention, comparator, food/drink type, outcome variable(s), outcome metric, sample size, age range, male–female ratio, length of study, country/ies where data collection took place, and results. Authors of the included articles were contacted to obtain the missing data.

### 2.4. Quality Assessment

Only peer-reviewed articles were included to ensure quality research. Included studies were assessed for risk of bias using an adapted version of the Cochrane tools, specifically RoB2 for randomized trials [35] and ROBINS-I for non-randomized trials [36]. Both tools assess the risk of four types of bias that can occur post-intervention: deviations from intended outcomes, missing outcome data, measurement of the outcome, and reporting of the result. For randomized studies, the tool also assesses bias due to the randomization process. For non-randomized studies, the tool assesses bias due to confounding, to the selection of participants, and to the classification of interventions.

### 2.5. Qualitative Synthesis

The studies were assessed in the context of age. We summarized and compared the studies conducted on children (3–12 years old), adolescents (12–18 years old), and adults (≥ 18 years old). Children’s cognitive skills are not fully developed, so their judgements and behaviors more strongly rely on visual cues compared to adolescents and adults. Moreover, they have lower levels of advertising literacy, which means that they are not (equally) capable of recognizing the persuasive intent of a message [13]. The studies were also assessed based on the type of cues investigated. We differentiated between visual and informational cues, and paid attention to whether they are present on packages of core or non-core food products. For reasons of brevity, only those cues are discussed, for which a sufficient number of studies were included. The results of the remaining cues can be found in the Appendix A.

## 3. Results

### 3.1. Study Characteristics

Fifty-seven studies in forty-three articles met our inclusion criteria (for an overview, see Appendix A). Nearly all the studies were conducted in high-income countries, including the USA, Belgium, Germany, the Netherlands, Canada, France, the UK, Australia, and Uruguay. Only two studies were conducted in what are considered upper-middle-income countries, specifically, Brazil and Guatemala [37].

Fifteen experiments were conducted exclusively on children (3–12 years old), thirty-nine experiments only investigated adults (≥ 18 years old), and three studies investigated both children and adults within a single experiment. No studies were conducted on adolescents (12–18 years old). In nearly all the studies, there was a balanced mix of male and female participants.

Front-of-pack cues were present on packages of one or more non-core food products (n = 24, e.g., sugary cereals, chocolate bars, pizzas, sugar-sweetened beverages), of one or more core food products (n = 6; e.g., fruit, vegetables, yoghurts, dried nuts, water), or of both core and non-core food products (n = 27). The main behavioral outcomes were product choices (n = 29), serving amounts (n = 5), and intake amounts (n = 23).

Forty-one studies took place in a controlled setting including a lab (n = 19), a laboratory store (n = 13), a school (n = 5), a cafeteria (n = 2), a workplace (n = 1), and a mock-up kitchen (n = 1). Of the thirteen studies that took place in a store setting, eight included eye-tracking measures and six included non-hypothetical choices where participants purchased or received at least one of the products they had selected. Thirteen studies took place in the field, including the people’s home (n = 5), a public place (n = 5), a shopping mall (n = 1), a club meeting room (n = 1), and a local cinema (n = 1). The three remaining studies were natural experiments, in which sales data were obtained from a supermarket or a cafeteria.

### 3.2. Study Quality Assessment

Among the randomized trials, the overall risk of bias was mostly moderate (n = 30), sometimes high (n = 13), and rarely low (n = 1). For many domains, studies lacked information to make thorough judgements about the level of bias. The risk of bias due to randomization, missing outcome data, and measurement of the outcome were considered low in most studies. Most studies failed to provide any information about blinding of the participants and research personnel, leading to moderate risks of intervention bias. Only few studies mentioned a pre-analysis plan, and most studies thus showed moderate risk of selection of the reported result bias. All non-randomized trials (n = 13) were judged to show moderate overall risk of bias. Four studies showed moderate risk of bias due to the baseline or time-varying confounding. The risk of bias due to selection of participants, classification of interventions, deviations from intended interventions, missing outcome data, and measurement of the outcome were considered low in most studies. A detailed overview of the quality assessment for all the studies can be found in the Appendix A.

### 3.3. Age

There were considerable differences between the types of interventions investigated in children and adults. All fifteen studies conducted exclusively on children investigated the impact of a visual front-of-pack cue, including endorsers (n = 8), illustrations (n = 4), branding elements (n = 2), and package size (n = 1). The impact of endorsers and illustrations was (almost) exclusively investigated on children. Twelve studies (i.e., 80.0%) concluded that visual cues influenced children’s food behavior. All studies but one found a direct effect of endorsers, all the studies found a direct effect of illustrations, and not one study found a direct effect of branding on children’s eating behaviors. The one study that investigated the impact of package size provides initial evidence that this cue influences children’s behaviors.

Only three studies investigated the impact of an informational cue (i.e., a label) on children’s behaviors. In all the three studies, a parent was also included in the sample. Only one study (i.e., 33.33%) found that labels influenced children’s product choices. Specifically, children and parents chose food products independently, and both made healthier choices when a simple nutrition label was present [38]. In another study, children and parents chose products based on packages with or without tasting the product. The presence of a nutrition or warning label did not have any impact on children’s food choices. The labels did influence adults’ food choices, but only when they did not taste the products [39]. The third study assessed whether children influence parents’ in-store food choices and found that nutrition labels did not lead child/parent duos to make healthier product choices [40].

Turning to adults, nineteen of the forty-two studies (45.23%) conducted on this age group investigated the impact of one or more visual cues. The visual cues differed greatly from those investigated on children. More specifically, cues included package size (n = 10), package shape (n = 3), product visibility (n = 2), resealability (n = 2), branding (n = 2), and illustrations (n = 1). One of these studies [41] investigated two different visual cues: package size and product visibility. The impact of package size, shape, product visibility, and resealability were (almost) exclusively investigated on adults. Fifteen studies (78.95%) concluded that visual cues influenced adults’ food behavior. All the studies found a direct effect of package shape, product visibility, and resealability, more than half the studies (i.e., 60.0%) found a direct effect of package size, and only one of the two studies (i.e., 50.0%) found a direct effect of branding on adults’ eating behaviors.

Twenty-five of the forty-two studies conducted on adults (i.e., 59.52%) investigated the impact of an informational cue (including the three previously mentioned studies that investigated both children and adults), more specifically, labels (n = 21), claims (n = 3), and congruence cues (n = 1). Two of these studies [42,43] investigated both a visual cue and an informational cue. Thirteen studies (52.0%) concluded that at least one informational cue directly influenced adults’ food behavior. There is mixed evidence concerning adults’ susceptibility to these cues. Ten studies (i.e., 47.62%) found a direct effect of a front-of-pack label and two studies (i.e., 66.66%) found a direct effect of a claim on adults’ eating behaviors. The one study that investigated the impact of a congruence cue suggests that this cue affects adults’ food behaviors.

One of the main research goals of this review is to compare the evidence on children and adults. Considering the great variability in the types of interventions investigated among these age groups, it is appropriate to examine these differences through a narrative review rather than a meta-analysis [44].

### 3.4. Cues

The studies included in this review investigated a range of visual and informational cues. In the section below, more details are provided on the effectiveness of these cues among children and adults (for an overview, see Table 1).

#### 3.4.1. Visual Cues

The majority of studies (n = 34) investigated the impact of one or more visual packaging cues on food choices, serving amounts or consumption amounts. Fifteen studies (i.e., 44.12%) were conducted on children, while the remaining nineteen were conducted on adults. The studies took place in varied settings, including labs, laboratory stores, public places, and people’s homes, with sample sizes ranging between 16 and 297. In 26 studies (i.e., 76.47%), the authors uncovered a main effect of at least one packaging cue.

##### Endorsers

Seven out of eight studies (i.e., 87.5%) found that endorsers directly affect children’s (age range: 3–12 years old) food behavior. In six studies, children were asked to choose a food package within one product category, with half of the packages portraying an endorser. All the studies concluded that children are more likely to choose a product that features an endorser, and this occurred for both core and non-core food products [46,47,48]. However, when asked to choose between products varying in healthiness, an endorser could not persuade children to choose a core food product over a non-core alternative. In the same article by Leonard et al. [46], one study that provided children with both core and non-core food products found that children consume more (ounces and calories) of the snack they like most, and that the expected food liking overrules the effect of an endorser. Two studies investigated effects on children’s food intake. Keller et al. [45] showed that the presence of an endorser influences children to eat more core foods, such as fruit and vegetables. Leonard et al. [46], however, found no effect of a licensed endorser on intake amounts of core or non-core foods.

##### Illustrations

All the studies found that illustrations affect eating behaviors of children (age range: 4–8 years old, n = 4) and adults (n = 1). In four studies, an illustration depicted a small or regular portion size compared to an exaggerated portion size. Children (n = 3) and adults (n = 1) poured and ate more food when the illustration depicted an exaggerated portion size [31,49,50]. This occurred for both core and non-core foods, except for the study by Aerts and Smits [31] where this effect only occurred for core foods. Macalister and Ethridge [51] investigated the impact of gender-specific illustrations on the food choices of 3–7 year-old children. Children were more likely to choose a package with gender-consistent packaging, and would even prefer a healthy snack with gender-consistent packaging over an unhealthy snack with gender-inconsistent packaging.

##### Package Size

Seven out of eleven studies (i.e., 63.64%) found that package size directly influences how much food people serve or eat. The studies were conducted on children (age range: 6–7 years old; n = 1) and adults (n = 10). The direction of this package size effect varied across studies. Two studies found that adults ate less core and non-core foods from a small package compared to a regular-sized package [55,56]. Haire and Raynor [54] found the same result, but only for adults who were overweight or obese. When small packages were compared to large packages, Eykelenboom et al. [53] found that adults did not serve themselves less non-core food from a smaller package, whereas van Kleef et al. [52] again found that people ate less from smaller packages. In four studies by Argo and White [41], multiple small packages were compared to a smaller number of large packages or no packages. Three studies found that a larger number of small packages actually increased intake, indicating a reversed effect of package size. One study failed to find an effect of package size. Finally, Aerts and Smits [16] provided evidence for a package size effect on young children, more specifically that children eat less non-core food from regular-sized packages compared to large packages.

##### Other visual cues

Studies assessed the impact of branding (n = 4), package shape (n = 3), product visibility (n = 2), and reresealability (n = 2) on food behaviors. The detailed results can be found in the Appendix A.

#### 3.4.2. Informational cues

Twenty-five studies investigated the impact of one or more informational cues on people’s food choices, serving amounts, or consumption amounts. All the studies were conducted on adults, and only three included children. The studies were carried out in varied settings, including labs, laboratory stores, and public places, with sample sizes ranging between 31 and 901 participants. In thirteen studies (i.e., 52.00%), the authors concluded that there was a main effect of at least one informational cue on a behavioral outcome.

##### Label

Twenty-one studies investigated the impact of one or more front-of-pack labels on product choices, intended intake, or actual intake. In ten studies (i.e., 47.62%), the authors concluded that a label had a direct positive impact on food behaviors. Most studies investigated the impact of a nutrition label. Two types of nutrition labels were differentiated: reductive labels and interpretative labels. Reductive labels provide factual information on the key nutrients such as calories, fat, sugar, and salt. Interpretative labels, on the other hand, help consumers deduct whether a nutrition score is good or bad, for instance, by adding a color scheme. Interpretative labels that are nutrient-specific provide information on different key nutrients, whereas summary interpretative labels provide an overall product score or rating [27].

Nutrient-specific interpretative label

Eighteen studies investigated the impact of one or more nutrient-specific interpretative labels including traffic light labels, Nutri-score labels, calorie labels, and warning labels. Only eight studies (i.e., 44.44%) found that this type of label triggered healthier food choices, and another eight studies found that the effect depended on characteristics such as age or weight status, or that the label only worked in combination with another informative element. All studies were conducted on adults, although three studies also included children.

Ten studies investigated the impact of a traffic light label (TFL) that evaluates key nutrient scores by allocating a color on a scale from green (good) to red (bad). Comparisons were made with packages containing a reductive label, another interpretative label, or no label. Only two studies (i.e., 20.0%) found that TFLs directly influenced healthier food choices. More specifically, packages with TFLs influenced adults to choose overall healthier products [66] and products with less sodium [68] compared to packages with no labels. Among the eight remaining studies, seven found an interaction effect. Graham et al. [40] found that TFLs influenced child/parent duos to make healthier choices, but only when there were additional signs that explained the labels. Lima et al. [39] found that TFLs influenced adults to choose a product with a slightly reduced sugar content, whereas nutrition warning labels influenced them to choose a product with highly reduced sugar content. However, both labels did not influence children’s product choices. Moreover, it was found that TFLs lead to healthier choices, but only among adults with low self-control [65] or when a label also includes a choice logo [64]. Sales data indicated that TFLs induced higher sales of ready-made meals (but not sales of sandwiches), but that this increase in sales was not associated with higher product healthiness [71]. Lastly, despite having contacted authors, there was one study for which the main effect of the traffic light label on adults’ product choices was unavailable [43].

One study investigated the impact of Nutri-score labels. These labels also use colors to evaluate the nutritional score, but does so for the overall nutritional quality of the product. Children and parents were asked to choose a food product for themselves and for each other. Fortunately, both children and adults consistently made healthier choices in the presence of a Nutri-score label [38].

Three out of four studies (i.e., 75.0%) found that calorie labels influenced adults to choose one or more products with fewer calories. Argo and White [41] compared packages with front-of-pack, back-of-pack, or absent calorie labels and concluded that only front-of-pack labels led to healthier choices. Newman et al. [74] investigated labels that combined calorie information with a star rating system and showed that these labels increased the likelihood of choosing a low-calorie product compared to a reductive label. Wegman et al. [70] investigated labels indicating whether a beverage was low or high in calories, and concluded that people were more likely to take home a beverage that was low in calories. However, another study on the impact of calorie labels on packages of sugar-sweetened beverages concluded that such labels did not lead to any changes in people’s drink choices [61].

Two out of four studies (i.e., 50.0%) found that nutrition warnings directly influenced food and drink choices of adults. Nutrition warnings that stated that a product was high or low in a key nutrient, such as sugar, sodium, or total fat influenced adults to avoid products with excessive amounts of these nutrients [73]. Warning labels that expressed general health consequences of excess sugar intake led adults to choose a beverage with less sugar compared to control labels [69]. Lima et al. [39] investigated the impact of traffic light labels and nutrition warnings on drink choices of both children and adults. As reported earlier, nutrition warnings influenced adults (but not children) to choose products with a highly reduced sugar content. Lastly, Mantzari et al. [61] investigated nutritional warnings to which fear appeal images were added, but failed to find any effect.

Summary interpretative label

Four studies investigated the impact of choice logos on packages of core food products indicating that the product is considered a healthy choice. Unfortunately, not one study concluded that choice logos had a direct effect on adults’ product choices or intake, although two studies did uncover an interaction effect. Scan data from a large panel of households indicated that logos induced an increase in the volume share of core food products, such as milk and yoghurt, but had no effect on the volume share of other products, such as cereals, fats, and oils [72]. In two other studies, the presence of a choice logo did not induce more core food choices compared to a traffic light label or to no label [64,66]. Nevertheless, as reported earlier, the combination of a traffic light label and a choice logo did lead to healthier choices [64]. Finally, in one study, adults were asked to serve and consume sugary cereals from a package with a choice logo, with a choice logo and information about calories per serving, or with no logo. Logos had no impact on the amount of food that adults’ poured or consumed for breakfast [62].

Serving size label

One out of two studies (i.e., 50.0%) found that serving size labels influenced adults’ food intake. Adults ate from small or large packages of a non-core food, where large packages could contain a label stating that the package contains either two or four servings. The labels directly influenced intake, such that people ate less when the product was labeled as four servings. Although adults ate more from large packages compared to smaller packages, this effect disappeared when the label mentioned that the package contained four servings [42]. In the study by Roberto et al. [48] (previously reported), food packages with a label containing a choice logo and the number of calories for different serving sizes had no impact on adults’ food choices.

##### Other Informational Cues

Three studies investigated whether one or more written claims impact the food behavior of adults. One study investigated whether information signaling that a product is (in)consistent with the consumption context influenced adults’ product choices. The detailed results can be found in the Appendix A.

### 3.5. Normative Influence

Twenty-six studies investigated the impact of cues that potentially function through normative influence. These studies included five types of cues, specifically, package size, illustrations, resealability, serving size labels, and endorsers. All the studies but three (i.e., 88.64%) concluded that cues influenced choices and behaviors related to both core and non-core products.

As reported earlier, people tend to eat more from large packages. Although this effect is often linked to consumption norms, no study actually measured whether the package size effect is mediated by changes in the perceived appropriate portion size. Several studies concluded that other cues, such as serving size labels and seals, can eliminate the package size effect. Two studies concluded that people eat less when a package provides the option for resealability [33]. One of these studies investigated whether this effect was mediated by a shift in perceived consumption norms. This was not the case, however, as the presence of a seal did not lead adults to think that it would be less appropriate to eat the complete package in one go. As with package size, increasing the displayed portion size influenced people’s intake, such that they ate more when the picture on the package demonstrated an exaggerated serving size.

One study investigated the impact of illustrations that were considered (in)consistent with a child’s gender, such as fairies and princesses for girls and cars and sunglasses for boys [51]. The authors concluded that children were more likely to choose foods with gender-consistent packaging, and hypothesized that this effect may be explained by social norms. Similar to adults, children want to fit in with their peers and seek out information about what is appropriate. However, young children are more motivated to fit into the gender expectations that they perceive. The illustrations may have impacted children’s food choices by creating such expectations.

## 4. Discussion

We conducted a systematic review (43 articles; 57 experimental studies) to summarize and compare the experimental evidence on the persuasiveness of front-of-pack cues to children and adults. The evidence suggests that package design can function as a cue that influences choices and eating behaviors. There is a similar amount of studies that supports an effect of visual front-of-pack cues on children (i.e., 80%) and adults (i.e., 73.7%), although there was a great variation in the types of cues investigated. Nearly all the studies on the impact of informational cues focus exclusively on adults. The evidence is mixed, with about half of the studies (i.e., 52%) supporting the effect of informational cues on adult food behavior. Our review only found three studies that investigated the impact of informational cues on food behaviors of children. Unfortunately, only one study found an effect, namely, that a simple nutrition label like the Nutri-score can influence children to make healthier choices. With new labels becoming more visual and easier to interpret (e.g., traffic lights or the Nutri-score), this urgently calls for research on how they affect children’s food behaviors.

Studies on children predominantly investigated visual marketing tactics, such as endorsers, illustrations, and brand elements. Children were influenced by packages containing licensed endorsers and (product) illustrations. This is a worrisome finding considering that these cues are still highly present on the packaging of non-core food products [17,18,31]. Children prefer products that feature a licensed endorser, but this cue cannot convince them to prefer a core over a non-core food product. This result corresponds with findings from the review by Smits et al. [22] that endorsers influence children’s behaviors, but that this influence is probably smaller for core food products. Illustrations could present a more promising tool to nudge children towards healthier eating behaviors. One study that investigated how children respond to gender (in)consistent illustrations concluded that children would rather choose a less tasty core food product with gender-consistent packaging than a non-core food product with gender-inconsistent packaging. This may be a valuable technique to induce healthier eating behaviors, yet difficult to implement in practice. Another important factor in influencing the children’s eating behavior is the depicted serving size. Packages that depict products with exaggerated serving sizes lead children to eat more, which is worrisome considering that this is often the case on the packaging of non-core food products. In comparison, packages that depict smaller (i.e., appropriate) serving sizes decreased intake, which suggests that this is a valuable technique to improve children’s eating behaviors. Another conclusion was that children’s food intake is not necessarily influenced by branding. While previous studies have concluded that children are aware of brands [7], that they find branded products tastier [79], and that children who are susceptible to brand images tend to eat more from meals that are branded [80], our review concludes that branding does not lead children to eat more (calories), although some effect may occur for girls and overweight children.

With regard to adults, their choices and behaviors are similarly influenced by front-of-pack cues, and more so by visual than by informational cues. There was some evidence in favor of a package size effect, specifically, that adults tend to eat more from large packages. Fortunately, many food packages have become smaller in recent years [81]. People eat more when they are given multiple small packages, however, a phenomenon referred to as the partitioning paradox [82]. Several studies outside this review have suggested that the impact of cues such as package and portion size can (partially) be explained by normative influence, whereby people rely on these cues to make judgements about the appropriate consumption size [83,84]. However, no study in the review measured whether perceived norms function as a mediator of the package size effect. Moreover, other cues, such as serving size labels and seals, can potentially eliminate or reduce the effect of package size. A serving size label that informs people that a package actually contains multiple servings can act as an additional source of normative information to correct judgements. The option to reseal the package also leads adults to consume less, although this effect could not be attributed to social norms.

There was less evidence to support the effect of informational cues, as only half of the studies (i.e., 52.00%) concluded that such a cue influences adults’ choices and eating behaviors. Overall, it seems that simple (i.e., interpretative) labels are more effective than detailed (i.e., reductive) labels. This finding may partly be explained by the fact that adults’ food behaviors are often quick and automatic [29], and reductive labels require more extensive cognitive processing [1,21]. Interpretative labels, such as calorie labels and nutritional warnings, are rather successful in inducing healthier product choices. Similarly, traffic light labels are common on food packages and considered a valuable tool to help consumers make out the nutritional value of a product [85]. Our review shows mixed evidence on the impact of TFLs, although, again, they are more effective than reductive labels. In recent years, Nutri-score labels have been introduced; they resemble traffic light labels, but are even more simplified and allow quicker interpretation. Despite the growing prevalence of this label on core and non-core food packages, only one study in our review investigated this cue. Results of this study suggest that this is a promising strategy to improve choice behaviors of both children and adults. There is one type of labels that should be especially easy for adults to interpret, i.e., choice logos. However, four studies that investigated choice logos found no effect on adults’ product choices. The findings from one study suggest that it may be valuable to integrate a logo with additional information, such as a traffic light color. Lastly, our review provides evidence that nutritional claims promote core product choices, whereas taste claims promote non-core product choices.

This review reveals avenues for future research. We did not find any studies on adolescents (12–18 years old), which indicates that this age group is heavily overlooked in the literature on food packaging. This lack of experimental studies on adolescents was also identified in reviews on the relationship between media food marketing and eating [10,15]. The review by Qutteina et al. [15] provides evidence that adolescents are susceptible to advertising in media channels. Adolescents have more buying power than children, and at the same time have low inhibitory control and impulsive decision-making [86]. Therefore, it is important to understand the extent and nature of their susceptibility to food packaging. Future studies should investigate the influence of both visual and informational cues on the eating behaviors of this age group.

Nearly all studies in the review were conducted in high-income countries, and only two studies were conducted in what are considered upper-middle-income countries [37]. Yet, childhood overweight and obesity is a major issue, especially in many low- and middle-income countries [87]. It is therefore important that more research on food packaging is conducted in low- and middle-income countries.

There was great variability in the types of cues investigated on children and adults, and the influence of many cues remains unexplored. Informational cues, such as nutrition claims, product claims, captions, and call-to-actions are prevalent on children-targeted food packages [17,19], especially of core foods [88]. And yet, our review hardly identified any studies that investigated the impact of these cues on children’s food behaviors. Future studies should also address the influence of visual cues, such as package size, shape, and promotions, on children-targeted food packages.

Many studies on adults investigated the impact of a nutrition label or claim. These cues often relate to other types of information, however, such as sustainability. There has been a rapid increase in the number of environmental schemes on food packages, such as eco-labels, footprint labels, and fair trade labels [89]. Despite the wide prevalence of sustainability labels and claims on food packages, this type of cues was not investigated in any of the studies in our review. We therefore call for more research on the impact of cues related to sustainability.

We realize that this review is not without limitations. First, we did not find any studies on adolescents, and thus our review cannot shed light on the persuasiveness of front-of-pack cues to the behaviors of this age group. Second, for a range of cues, there was only a limited amount of evidence, which means that we cannot draw firm conclusions about the impact of these cues. Finally, the studies predominantly investigated short-term effects of package cues on eating behaviors measured at one fixed time point. Little is known about whether the effect of these cues persists in the long term. There were a few studies that measured food intake over a longer period of time, and these provide initial evidence in favor of a long-term effect.

To our knowledge, this is the first review on the impact of front-of-pack cues on product choices and eating behaviors of children vs. adults. The current review provides a holistic overview of the impact of different packaging cues on the eating behavior, and sets itself apart from reviews that focused on only one type of cues. The findings of our review emphasize the importance of initiatives to limit or control the market’s use of visual packaging cues, especially those targeted at children. There have been initiatives to restrict marketing of unhealthy foods to children via media channels, but, unfortunately, the initiatives to restrict marketing on food packages are still lacking. Our review also demonstrates the potential of visual cues to induce healthier behaviors in children and adults. Policymakers do good to enforce informational cues such as labels, but not all labels are effective, and our findings indicate they should be simple and easy to understand.

## Figures and Tables

**Figure 1 nutrients-12-01062-f001:**
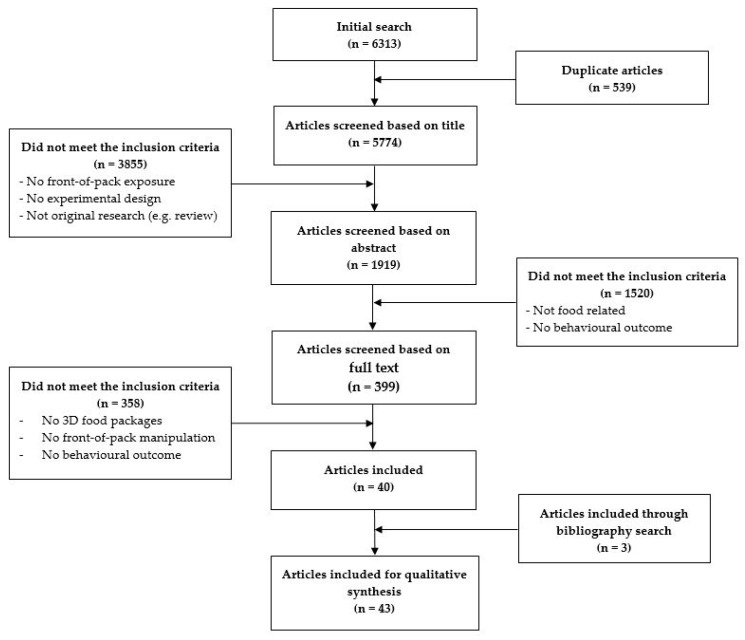
Flowchart depicting the article screening process.

**Table 1 nutrients-12-01062-t001:** Overview of front-of-pack cues across the included studies.

Category	Type of Cues	Intervention	Main Outcome(s) Measured	Number of Studies	Overall Results ^1^
Visual cue	Endorser [45,46,47,48]	Licensed endorser	Choice; Intake	n = 8 Children (100%)/Adults	Sample size range: 16–139 Main effect in n = 7/8, i.e., 87.5%
	Illustration [31,49,50,51]	Product picture; gender-(in)consistent image	Choice; Serving; Intake	n = 5 Children (80%)/Adults	Sample size range: 22–212 Main effect in n = 5/5, i.e., 100.0%
	Package size [16,41,52,53,54,55,56]	Small/regular/large packaging	Serving; Intake	n = 11 Children (0%)/Adults	Sample size range: 19–297 Main effect in n = 7/11, i.e., 63.6% Interaction effect in n = 1
	Package shape [57]	Squeeze-ability	Serving	n = 3 Children (0%)/Adults	Sample size range: 71–108 Main effect in n = 3/3, i.e., 100.0%
	Resealability [33]	Presence of a seal	Intake	n = 2 Children (0%)/Adults)	Sample size range: 43–79 Main effect in n = 2/2, i.e., 100.0%
	Branding [43,45,58,59]	Logo; branded packaging	Choice; Intake	n = 4 Children (50%)/Adults	Sample size range: 41–104 Main effect in n = 1/4, i.e., 25.0% Interaction effect in n = 2
	Product visibility [41,60]	Packaging partially/completely covered	Choice; Intake	n = 2 Children (0%)/Adults	Sample size range: 28–207 Main effect in n = 2/2, i.e., 100.0%
					Total: Main effect in 27/35 ^2^, i.e., 77.14%
Informational cue	Label [38,39,40,41,42,43,57,61,62,63,64,65,66,67,68,69,70,71,72,73,74]	Traffic light; Nutri-score; calories; nutritional warning; choice logo; serving size	Choice; Serving; Intake	n = 21 Children (14.29%)/Adults (100.0%)	Sample size range: 28–7216 households Main effect in n = 1021, i.e., 47.62% Interaction effect in n = 10
	Claim [75,76,77]	Nutrition claim; health claim; taste claim	Choice; Intake	n = 3 Children (0%)/Adults	Sample size range: 135–210 Main effect in n = 2/3, i.e., 66.66% Interaction effect in n = 1
	Congruency cue [78]	Appropriateness within the product category	Choice	n = 1 Children (0%)/Adults	Sample size: 168 Main effect in n = 1/1, i.e., 100.0%
					Total: Main effect in 13/25, i.e., 52.0%

^1^ Main effect is considered significant at *p* < 0.05. The answer is “Yes” if a study found a significant main effect of at least one front-of-pack cue. ^2^ Total number of studies investigating a visual cue is n = 34, but one study investigated two different types of visual cue.

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
