# Peer review of "That’s My Cue to Eat: A Systematic Review of the Persuasiveness of Front-of-Pack Cues on Food Packages for Children vs. Adults"

_nutrients, 2020, doi:10.3390/nu12041062_

Round 1
Reviewer 1 Report
The study of Hallez L. et al., is a well written and designed systematic review investigating the effects of all possible fron-of-pack cues on choices abd food behaviors. Moreover, the value of their study increases as they collect experimental evidence of different age groups.
The methodology of their study is well explained and is based on the preestablished existing criteria of systematic reviews. The selection of the studies is adequate and is worth mentioning that authors have studied in depth the selected studies.
Moreover, they have been able to classify the results of the studies therefore achieving the objective of their study. They have been able to provide a holistic review of the parameters that may influence the buying choices of the public.
The results of the present study can help in the design of future nutritional strategies and campaigns.
Author Response
Dear reviewer
Thank you for the swift and positive feedback with regards to our manuscript.
Lotte Hallez
(in name of all authors)
Reviewer 2 Report
This paper is a narrative review of studies with behavioral and/or intake outcomes related to food packaging. Overall, it is well written and provides a succinct overview of the current state of food packaging and its effects on adults' and children's' behaviors. I feel that it would be of interest to the journal’s audience. I have just a few minor comments.
Lines 36-38: Is there a citation for this statement?
Lines 62-63: Should read “A considerable number of studies have”
Lines 68-69: Should read “Similarly, a growing number of studies have”
Lines 81-82: Is there a citation for this statement?
Lines 197-198 It's not clear what, “and in that case a parent”, means. Do you mean in 1 of the 3 studies or in all of the 3 studies?
Lines 438-439: Has this statement been tested or is this speculation. If speculation it should be suggested as a future direction.
Lines 440-443: Additionally, there is a paper (https://doi.org/10.1007/s11682-018-9919-8) that shows that children who are susceptible to brand images (both food and non-food; as measured by fMRI) eat more in the presence of brands.
Author Response
Response to Reviewer 2 comments
Dear reviewer
Thank you for the swift and valuable feedback with regards to our manuscript. We have made minor revisions based on your feedback. A new version of the manuscript will be uploaded including track changes.
We would like to present the following responses to your comments:
Point 1. Lines 36-38: Is there a citation for this statement?
Lines 36-38 in revised manuscript: “Packages also constitute an element of the food environment at the moment of consumption and can influence decisions on what to eat and how much to eat”. A citation was added to this statement (http://dx.doi.org/10.2139/ssrn.2083618). This review by Chandon demonstrates the impact of package design elements (e.g. package size, shape, etc.) on consumption.
Point 2. Lines 62-63: Should read “A considerable number of studies have”
Line 62 in revised manuscript: Revised as requested.
Point 3. Lines 68-69: Should read “Similarly, a growing number of studies have”
Line 68 in revised manuscript: Revised as requested.
Point 4. Lines 81-82: Is there a citation for this statement?
Lines 81-83 in revised manuscript: “Similarly, product pictures can create expectations about appropriate serving sizes, such that pictures where the portion size is exaggerated can lead to higher intakes”. Two citations were added to this statement (https://doi.org/10.1111/ijcs.12503; https://doi.org/10.1016/j.appet.2014.06.018). The paper by Aerts and Smits demonstrates that the portion size depicted on packages influences consumption. The paper by Marchiori; Papies; Klein provides evidence for an underlying mechanisms of this portion size effect, namely that portion size functions as an anchor for people’s judgements about appropriate serving sizes. Lines 81-83 were slightly rewritten based on the information from these citations.
Point 5. Lines 197-198 It's not clear what, “and in that case a parent”, means. Do you mean in 1 of the 3 studies or in all of the 3 studies?
Lines 198-199 in revised manuscript: “Only three studies investigated the impact of an informational cue (i.e. a label) on children’s behaviors, and in that case a parent was also included in the sample”. Line 199 was rewritten to clarify that a parent was included in the sample in all three studies.
Point 6. Lines 438-439: Has this statement been tested or is this speculation. If speculation it should be suggested as a future direction.
Lines 438-441 in revised manuscript: “Therefore, a valuable technique to improve children’s eating behaviors is to depict product pictures with smaller serving sizes”. This statement has been tested in different studies included in the review and builds on the findings of these papers (https://doi.org/10.1016/j.appet.2015.07.003 ; https://doi.org/10.1111/ijcs.12503). The paper by Neyens; Aerts; Smits demonstrates that young children eat less cereal when the product picture portrays a small vs. large portion size. The paper by Aerts; Smits also demonstrates in two studies that children eat less when the picture portrays a regular (i.e. appropriate) vs. a large portion size. Lines 439 - 442 were rewritten to make this more clear.
Point 7. Lines 440-443: Additionally, there is a paper (https://doi.org/10.1007/s11682-018-9919-8) that shows that children who are susceptible to brand images (both food and non-food; as measured by fMRI) eat more in the presence of brands.
Line 443-447 in revised manuscript: “While previous studies have concluded that children are aware of brands and find branded products tastier, our review concludes that branding does not lead children to eat more (calories), although some effect may occur for girls and children with overweight”. A statement (+ citation) has been added to this, as the paper recommended by the reviewer (https://doi.org/10.1007/s11682-018-9919-8) contributes to the evidence on the link between branding and children’s cognitive and behavioral responses.